# No Need to Know Everything!
# Efficiently Augmenting Language Models With External Knowledge

**Jivat Neet Kaur**[1]*                                          JIVATNEET@GMAIL.COM
**Sumit Bhatia**[2]                                          SUMIT.BHATIA@ADOBE.COM
**Milan Aggarwal**[2]                                          MILAGGAR@ADOBE.COM
**Rachit Bansal**[3]*                                          RACBANSA@GMAIL.COM
**Balaji Krishnamurthy**[2]                                          KBALAJI@ADOBE.COM

[1]*Microsoft Research India*
[2]*Media and Data Science Research Lab, Adobe*
[3]*Delhi Technological University*

## Abstract

Large transformer-based pre-trained language models have achieved impressive performance on a variety of knowledge-intensive tasks and can capture semantic, syntactic, and factual knowledge in their parameters. However, storing large amounts of factual knowledge in the parameters of the model is sub-optimal given the resource requirements and ever-growing amounts of knowledge. Instead of packing all the knowledge in the model parameters, we argue that a more efficient alternative is to provide contextually relevant structured knowledge to the model and train it to use that knowledge. This allows the training of the language model to be de-coupled from the external knowledge source and the latter can be updated without affecting the parameters of the language model. Empirical evaluation using different subsets of LAMA probe reveals that such an approach allows smaller language models with access to external knowledge to achieve significant and robust outperformance over much larger language models.

## 1. Introduction

Large pre-trained language models (PLMs) [Peters et al., 2018, Devlin et al., 2019, Brown et al., 2020] have achieved state-of-the-art performance on a variety of downstream NLP tasks. Much of this success can be attributed to the significant semantic and syntactic information captured in the contextual representations learned by PLMs. In addition to applications requiring linguistic knowledge, PLMs have been useful for a variety of tasks involving factual knowledge and it has been shown that models such as BERT [Devlin et al., 2019] and T5 [Raffel et al., 2020] store significant world knowledge in their parameters [Petroni et al., 2019]. For instance, BERT can successfully predict the masked token in this cloze sentence, "`[MASK] is the capital of Australia`" as `Canberra`. However, the knowledge captured by these PLMs is stored implicitly in the form of parameter weights making it difficult to explicitly retrieve the knowledge, and even to quantify how much knowledge is stored in the models. Further, PLMs do not contain explicit grounding to real world entities, and hence, often find it difficult to recall factual knwoledge [Logan et al., 2019]. Moreover, since the PLMs acquire knowledge from the text corpora they are trained on, they tend to become sensitive to the context and linguistic variations [Jiang et al., 2020]. For example, the model may not be able to recall correct information and successfully complete the

---

*work done during summer internship at Adobe

sentence, "*The birthplace of Barack Obama is* ___", if the LM has seen this fact in a different context during training (e.g., "*Barack Obama was born in Honolulu, Hawaii.*"). This not only presents a lack of interpretability while analyzing model predictions but also poses constraints on the amount of knowledge that can be stored. Capturing greater world knowledge requires training ever-larger resource-hungry networks that have great financial and environmental costs.

In this work, we step back and ask – what if instead of focusing on storing the knowledge in language model parameters, we provide the model with contextually relevant knowledge and train it to use this knowledge. This approach has several advantages – *(i)* we can utilize the already available abundant large-scale knowledge bases such as Yago [Suchanek et al., 2007] and Wikidata [Vrandečić and Krötzsch, 2014]; *(ii)* not all the knowledge needs to be packed in the parameters of the model resulting in lighter, smaller and greener models; and *(iii)* as new knowledge becomes available, the knowledge base can be updated independently of the language model.

**Differences from Prior Work:** Previous work on augmenting PLMs with additional knowledge can be grouped into two categories. One line of work has focused on injecting the knowledge directly into the model parameters by feeding more data to the model during pre-training [Zhang et al., 2019, Peters et al., 2019, Poerner et al., 2020, Roberts et al., 2020]. However, these approaches, by definition, will lead to larger and larger models to store the ever-growing abundant knowledge. In addition, updating or adding new knowledge in the model requires further training of the model.

The second body of work adopts a retrieve and read framework where the model is trained to retrieve relevant information followed by a reading comprehension step to perform the downstream task [Lee et al., 2019, Guu et al., 2020, Agarwal et al., 2021]. Our proposal is closely aligned with this body of work but with two main differences. First, most of these existing works have considered external knowledge in the form of unstructured text (such as Wikipedia documents and factual verbalized statements). However, extracting factual knowledge from unstructured text is hard and error-prone due to ambiguities in natural language and infrequent mentions of entities of interest [Peters et al., 2019]. For example, in a typical text corpus such as Wikipedia, the date of birth of a famous personality (say Barack Obama) will have many more occurrences compared to facts about lesser-known persons (say authors of this paper). This issue can be alleviated by using a structured knowledge base (such as Yago) where the knowledge is represented unambiguously – each fact is a triple in the knowledge base. Further, the existing approaches for retrieval employ supervision during pre-training to train the model to fetch relevant information. This results in systems that are more complex and resource-hungry than the base PLMs used and also make it difficult to reuse or adapt the models to different sources of knowledge. Inspired from the recent findings of Petroni et al. [2020] that have shown that such complex supervision may not be necessary, we devise a simple, yet efficient, way of retrieving contextually relevant structured knowledge.

**Our Contributions:** In sum, we present an approach for augmenting PLMs with contextually relevant structured knowledge (Section 2) and find that it leads to significant performance improvements as measured by popular knowledge probes (Section 3). Specifically, we show that with access to contextually relevant knowledge, the much smaller BERT-base model significantly outperforms the much larger BERT-large variant and is less sensitive to contextual variations in input. We also present various examples of cases where the addition of knowledge helps the model and also present an in-depth analysis of cases where the model gives incorrect answers (Section 3.4).

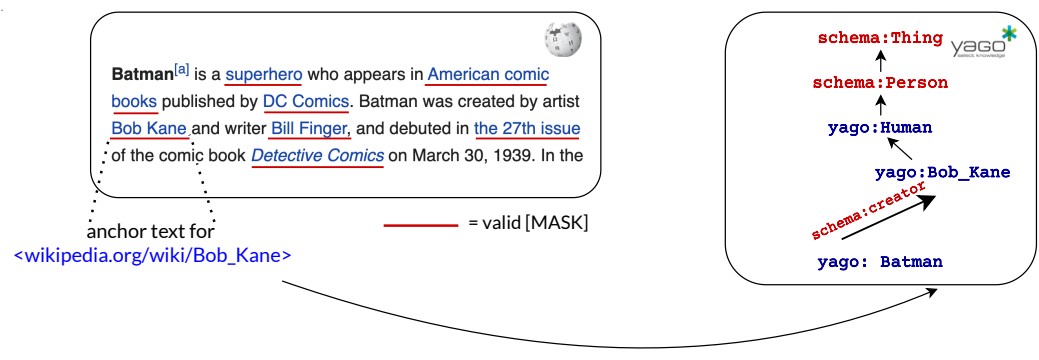

Figure 1: We create our pre-training corpus from Wikipedia by masking entity spans detected using anchor text of hyperlinks (details in Section 2.1). During pre-training, we randomly sample a mask from the sentence to create our input.

## 2. Augmenting PLMs With External Knowledge

PLMs are typically fed a large amount of unstructured text and the linguistic nuances and world knowledge are captured in the model parameters. Since it is not practical to pack all the world knowledge in the parameters of the language model, we aim to develop an approach where the language model has access to an external knowledge source and is instead trained to utilize this knowledge on demand. Such a system would not require all the knowledge to be represented and stored in the model parameters. We posit that the model essentially needs to learn associations between natural language input text and various relation types present in the knowledge base. Identifying the correct relation types will help the model leverage the corresponding relevant facts in order to make an accurate prediction. We achieve this objective via a modified MLM pre-training objective and our proposed system architecture is summarized in Figure 2. There are three main components in our proposed approach. First, we describe our novel approach to create masks (Section 2.1). Next, we explain our strategy to retrieve contextually relevant knowledge from the underlying knowledge base (Section 2.2), and finally, we elaborate on our pre-training approach (Section 2.3).

### 2.1 Entity span masking

Masked Language Modelling (MLM) [Devlin et al., 2019] is a popular task used for training large language models where the objective is to predict the masked token in the input sequence. In order to improve model's grounding to real world entities, previous works have adopted different strategies for explicitly masking entity information in the input text by using entity representations obtained by knowledge base embeddings [Zhang et al., 2019], using named entity taggers and regular expressions [Guu et al., 2020], and verbalizing knowledge base triples [Agarwal et al., 2021]. These approaches often result in noisy masks due to the limitations of underlying rules, and NER and entity linking systems.

We propose a novel way of creating high-quality and accurate entity masks for training using the MLM objective. Using Wikipedia as the base corpus for training and Yago as our knowledge base, our approach of creating the training data is summarized in Figure 1. In order to create entity masks, we need to identify corresponding entity mentions in the input text for which we utilize the

human-annotated links in Wikipedia to identify entity tokens and their corresponding entities in the knowledge base. Wikipedia style guidelines require editors to link mentions of topics to their corresponding Wikipedia pages. In Figure 1, the left textbox shows a screenshot of the Wikipedia article about `Batman` where other related topics, or concepts, are linked to their corresponding Wikipedia pages (underlined in red in the figure, and displayed as blue anchor-text in Wikipedia). This information provides us with high-quality human annotation of entity mentions in the input text. Further, we used the English Wikipedia instance of Yago that only contains entity instances having a corresponding English Wikipedia article. We utilize this direct mapping between Yago entities and Wikipedia articles to create our training data. As illustrated in Figure 1, the underlined tokens (such as `DC Comics, Bob Kane, Bill Finger`) constitute the set of entity tokens that could be masked. For each such mask, we have the corresponding mapping in the Yago knowledge base (illustrated for `Bob Kane` in the right text box). By masking only the entity tokens (instead of randomly sampled words) and providing the contextually relevant knowledge to the model (described in next subsection), we expect the model to learn to predict the masked entity tokens by utilizing the external knowledge.

## 2.2 Contextual Knowledge Retrieval

After preparing the masked input for training as described above, the second component in our solution fetches contextually relevant knowledge to feed to the language model along with the input. Consider the input sentence, "*Warren Buffet is the chairman of [MASK]*", where the masked token is *Berkshire Hathaway*. In the typical MLM setting, the model only has access to the linguistic and contextual clues present in the input text to predict the masked token. However, if contextually relevant information is available as additional input, the model can use it to output the correct masked token. Previous works have tried providing this additional knowledge in the form of verbalized triples [Agarwal et al., 2021] that stores the knowledge in the model parameters, by feeding additional relevant documents in a retrieve and read framework [Guu et al., 2020], or by learning joint-representations of language and entities in knowledge base [Yasunaga et al., 2021]. However, such approaches require heavy computational resources or are designed for a specific downstream task. On the other hand, we aim to develop a lightweight retrieval component that can fetch highly relevant knowledge on-demand which is then passed on to the language model. Such an approach, while being simple, requires minimal computational overhead and allows for the external knowledge to be updated with no impact on the language model parameters.

We consider the problem of finding contextually relevant facts from the knowledge base given the input query as an information retrieval (IR) problem and adopt a retrieve and re-rank approach that has empirically been found to perform well in a variety of tasks [Chen et al., 2017, Wang et al., 2017, Das et al., 2019, Yang et al., 2019]. Recall the example input discussed above – "*Warren Buffett is the chairman of [MASK]*". Intuitively, in this input text, there are two important signals that the retriever needs to utilize – *entity* and *relation* information. First, the entity mention *Warren Buffett* indicates that we need to fetch facts related to Warren Buffet from the knowledge base. Typically, there are numerous facts related to a given entity in the knowledge base, especially for popular entities such as `Warren Buffett`. Thus, the retriever also needs to utilize the presence of the word chairman to weigh facts (knowledge base triples) representing the management or executive relation.

Given an input text, our retriever pipeline performs Named Entity Recognition (NER) to identify named entity mentions in the input text. We use the NER model from FLAIR [Akbik et al., 2019] to identify named entity mentions and then select KB entities that have maximum overlap with the mention-span of the identified named entities. For instance, if the input query is "*Buffett was born in [MASK]*", all entities containing Buffett – `Warren_Buffett`, `Howard_Warren_Buffett`, `Howard_Graham_Buffett`, `Volcano_(Jimmy_ Buffett_song)` etc. are selected, but if the query is "*Warren Buffett was born in [MASK]*", only the first two entities will be chosen). Once these entities are selected, all the facts from the knowledge base (triples) involving these entities are retrieved (denoted by $\mathcal{T}_x$ in Figure 2).

After retrieving the facts involving the entities mentioned in the input, we next need to rank these triples based on their relevance to the input. In order to measure the contextual relevance of a given triple $t$ to the input $x$, we compute the following two scores.

1. **Query-Triple similarity:** We obtain representations of the input text $x$ as well as the triple $t$ and compute the inner product of the representations to obtain the similarity score as follows.

$$sim(x,t) = \texttt{Embedding}(x)^T \texttt{Embedding}(t), \ t \in \mathcal{T}_x \qquad (1)$$

   Here, `Embedding`$(\cdot)$ is obtained using the Sentence Transformer [Reimers and Gurevych, 2019]. While it is straightforward to obtain representations of input $x$, sentence transformer can not be applied directly to knowledge base triples. Application of knowledge base embeddings such as TransE [Bordes et al., 2013] is also not feasible as then the representations of the input text and triples will be in different embedding spaces. To overcome this, we adopt a simple approach of verbalizing the knowledge base triples by concatenating the head entity, relationship and the tail entity, and then obtain the representation of the verbalized triple from the sentence transformer. For example, the triple (`Warren_Buffett`, `hasOccupation`, `Investor`) is verbalized as *Warren Buffett has occupation Investor* and is fed as input to the sentence transformer.

2. **Relation-based scoring:** A triple is highly relevant for the input text if the triple represents the same relationship that is being talked about in the text. To capture this intuition, we embed all the relation types in Yago in the same embedding space as triples using the sentence transformer and compute the similarity between the input text and the relation type of the triple as follows.

$$sim(x,r) = \texttt{Embedding}(x)^T \texttt{Embedding}(r), \ r \in \mathcal{R} \qquad (2)$$

   where $\mathcal{R}$ is the set of all relations in the KB.

The final relevance score for the triple $t$, $relevance(x,t)$ is obtained by taking a product of the above two scores.

$$relevance(x,t) = sim(x,t) \times sim(x,r_t) \qquad (3)$$

Based on $relevance(x,t)$, we select the top-$k$ triples that constitute the contextual knowledge to be fed as input along with $x$ to the LM. We use $k = 8$ in this work. Some illustrative examples of the final retrieved knowledge base triples are presented in Table 1.

| Input | Masked Token | Candidates |
|---|---|---|
| Henri Jules Louis Marie Rendu (24 July 1844 – 16 April 1902) was a French physician born in [MASK]. | Paris | (Henri Jules Louis Marie Rendu; birth date; 1844-07-24) **(Henri Jules Louis Marie Rendu; birth place; Paris)** (Henri Jules Louis Marie Rendu; death date; 1902-04-16) (Henri Jules Louis Marie Rendu; nationalit;y France) (Henri Jules Louis Marie Rendu; given name; Henri) |
| Weisenborn attended the [MASK]. | University of Chicago | **(Gordon Weisenborn; alumni of; University of Chicago)** (Clara Weisenborn; member of; Republican Party (United States)) (Günther Weisenborn; nationality; Germany) (Günther Weisenborn; death place; West Berlin) (Clara Weisenborn; nationality; United States) |
| Dehorokkhi (English: Bodyguard) is a Bangladeshi [MASK] directed by Iftakar Chowdhury. | action film | (Dehorokkhi; director; Iftakar Chowdhury) (Dehorokkhi; in language; Bengali language) **(Dehorokkhi; genre; Action film)** (Bangladeshi Idol; in language; Bengali language) (British Bangladeshi Who's Who; in language; English language) |
| Palaemon macrodactylus is a [MASK] of shrimp of the family Palaemonidae. | species | (Palaemon macrodactylus; parent taxon Palaemon (genus)) (Palaemon macrodactylus; parent taxon; Palaemon (genus)) **(Palaemon macrodactylus; taxonomic rank; Species)** (Palaemonidae; taxonomic rank; Family (biology)) (Palaemonidae; parent taxon; Palaemonoidea) |

Table 1: Examples of masked input sentences (from Wikipedia) and top-5 retrieved candidates during pre-training (candidates containing correct fact in bold).

## 2.3 Language Model Pre-training with Contextual Knowledge

With the masked corpus and the module to fetch contextually relevant knowledge, we now train the model to utilize the additional contextual knowledge to predict the masked token. From the masked corpus, we select a sentence and a valid entity span out of all the potential spans in the sentence is chosen at random and masked to create the input text $x$. We filter out sentences starting with pronouns such as *he, she, her*, and *they* as we observed that most of such sentences do not contain other useful signals to unambiguously predict the masked words. For instance, if the input example is - "*He developed an interest in investing in his youth, eventually entering the Wharton School of the University of Pennsylvania*" and *Wharton School of the University of Pennsylvania* is masked, the remaining sentence ("*He developed an interest in investing in his youth, eventually entering*") is not providing any informative signals to the model to predict *Wharton School of the University of Pennsylvania*.

Given the input sentence thus selected, the contextual knowledge retriever fetches the relevant triples from the knowledge base. The representations of the input sentence and the retrieved triples are then concatenated and fed to the model and the model is trained to minimize the following MLM loss.

$$L_{MLM} = \frac{1}{M} \sum_{m=1}^{M} \log p(x_{ind_m} \mid x, t_1, t_2, ..., t_k) \tag{4}$$

where $M$ is the total number of [MASK] tokens in $x$ and $ind_m$ is the index of the $m^{th}$ masked token.

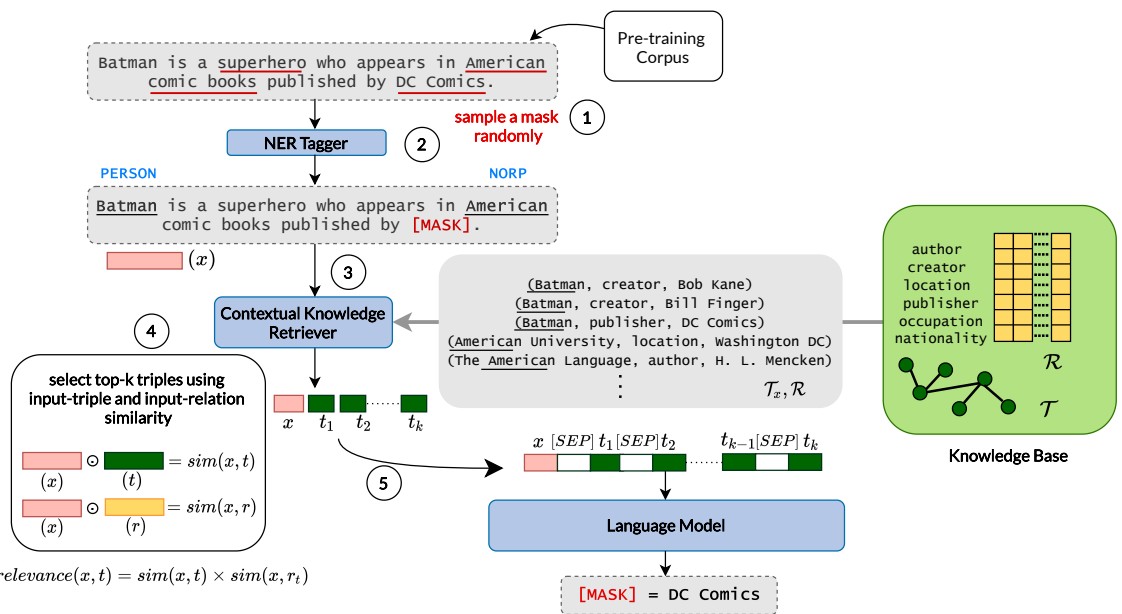

Figure 2: **Augmenting Language Model Pre-Training with Contextual Knowledge.:** ①Using a sentence sampled from the pre-training corpus, an input $(x)$ is created by selecting an entity mention at random from the potential mask candidates (underlined in red). ②An NER tagger is then applied to the masked input sequence $(x)$ to identify named entities (underlined in black). ③For the identified entities, the Knowledge Retrieval module fetches the set $\mathcal{T}_x$ of all the triples from the Knowledge Base and then ④scores all the retrieved triples using input-triple and input-relation similarity (details in Section 2.2). ⑤The top-$k$ triples are fed to the Language Model encoder along with the input sequence $(x)$ and the model is trained to predict the masked token.

With the additional contextual information available to the model, we expect the model to learn the associations between the linguistic cues in the input text, and the relevant relationship information in the triples. For example, we expect the model to associate different ways in which someone's date of birth could be mentioned in natural language (such as *X was born on*, *the birthday of X is*, and numerous other linguistic variations) to the KB relation `birthDate` and utilize the information from the corresponding triple. Note that since the types of relations in the knowledge base are relatively small in number, and do not change often, we expect the model to generalize well and be more robust to linguistic variations.

## 3. Empirical Evaluation and Discussions

### 3.1 Data Sources and Pre-processing

We create our pre-training corpus using the December 20, 2018 snapshot of English Wikipedia that contains about 5.5M documents. Processing all the articles following the masking strategy described in Section 2.1 resulted in a total of ∼46.3M sentences with valid masks, from which we randomly

|  | Google-RE | | | | T-REx | | | | SQuAD | Concept Net |
|---|---|---|---|---|---|---|---|---|---|---|
|  | DoB | PoB | PoD | Total | 1-1 | N-1 | N-M | Total |  |  |
| **Elmo 5.5B** | 0.10 | 7.50 | 1.30 | 3.00 | 13.10 | 6.50 | 7.40 | 7.10 | 4.30 | 6.20 |
| **Tranformer-XL** | 0.90 | 1.10 | 2.70 | 1.60 | 36.50 | 18.00 | 16.50 | 18.30 | 3.90 | 5.70 |
| **BERT-base** | 1.59 | 15.46 | 10.33 | 9.12 | 67.94 | 32.67 | 23.54 | 30.83 | 14.29 | 15.88 |
| **BERT-large** | 1.59 | 15.53 | 12.16 | 9.76 | 74.23 | 31.30 | 25.30 | 31.05 | 17.61 | **18.72** |
| **Ours (BERT-base)** | **64.44** | 52.71 | 50.98 | 56.04 | **74.37** | 51.18 | 34.57 | 45.83 | 15.61 | 14.78 |
| **REALM** | 49.06 | **79.56** | **64.13** | **67.36** | 55.81 | **69.54** | **66.98** | **68.18** | **27.96** | 4.78 |

Table 2: Mean precision at one (P@1) of various models on LAMA probe. Best results are high-lighted in **bold** and the second best performance is underlined.

sample sentences to create input examples. Our LM encoder is the uncased BERT-base model (12 layers, 768 hidden units, 12 attention heads).

For our knowledge base, we use the English Wikipedia version of Yago 4 [Suchanek et al., 2007] [1]. We pre-process the Yago4 knowledge base to remove triples involving relationships such as *image, logo* and *url* that point to meta-data such as images and other files, and triples that point to RDF literals or Wikidata URLs. This results in the final knowledge base consisting of roughly 17M triples spanning over 131 unique relations. For computing triple representations for retrieval (Section 2.2), we concatenate the subject (head), relation, and object (tail) of triples and embed them using the Sentence Transformers [Reimers and Gurevych, 2019] and obtain the 768-dimensional embeddings (same as LM encoder dimensions).

### 3.2 Does External Knowledge Help PLMs in Knowledge Intensive Tasks?

We now present an analysis of how much, and if, having access to external knowledge can help PLMs in knowledge-intensive tasks. A popular way of assessing a model's ability to perform at such tasks is by using benchmark knowledge probes. We use the LAMA probe [Petroni et al., 2019] – a comprehensive knowledge probe designed to tests for various types of knowledge by combining facts from four different types of knowledge sources - Google-RE[2], T-REx [Elsahar et al., 2018], SQuAD [Rajpurkar et al., 2016] and ConceptNet [Speer et al., 2017a]. It provides a cloze-style representation of facts and the model being evaluated is required to predict the masked words in these sentences (e.g., *Barack Obama was born in ___ .*).

Table 2 reports the performance of various PLMs on the LAMA probe. We report results for Elmo [Peters et al., 2018], Transformer-XL [Dai et al., 2019], BERT base and large variants [Devlin et al., 2019] and our proposed approach with BERT-base as the language encoder and Yago4 as the external knowledge base. We observe that our approach of providing external knowledge to the PLMs leads to substantially improved performance, especially for Google-RE and T-REx subsets of the probe. Compared to the 9.76% mean precision at one (P@1) achieved by BERT-large on Google-RE, our solution (built using the much smaller BERT-base) achieves an impressive precision of 56.04%. Similarly, on the T-REx dataset, the proposed approach achieves 45.83% mean P@1 outperforming BERT-large (31.05%). We do note that on the SQuAD and Concept Net sub-

---

[1]https://yago-knowledge.org/downloads/yago-4
[2]https://code.google.com/archive/p/relation-extraction-corpus/

|          | BERT-base | BERT-large | Ours (BERT-base + Yago4) |
|----------|-----------|------------|--------------------------|
| LAMA     | 26.93     | 27.01      | **47.06**                |
| LAMA-UHN | 17.49     | 18.38      | **40.40**                |
| % drop   | 35.05     | 31.95      | **14.15**                |

Table 4: Precision@1 on LAMA-UHN for various models.

sets, the proposed solution is not able to outperform BERT-large, though the performance of the proposed solution is close to BERT-base. We attribute this difference in performance to the nature of probes in the four subsets. While Google-RE and T-REx focus more on probing factual world knowledge (present in abundance in Yago), SQuAD and Concept Net concentrate more on commonsense knowledge (limited in Yago). This is a major focus of our continuing work on enhancing the external knowledge with commonsense knowledge bases such as Atomic [Sap et al., 2019] and ConceptNet [Speer et al., 2017b].

Table 2 also reports results of REALM [Guu et al., 2020] – a retrieval-based language model that retrieves relevant documents from a text corpus during pre-training. We observe that the proposed solution outperforms REALM on the ConceptNet, DoB (Google-RE), and 1-1 (T-REx) subsets, while REALM outperforms the proposed solution in other subsets of the LAMA probe. We specifically highlight an absolute 15 points improvement on the `date-of-birth` relation as REALM uses explicit date masks while training whereas our training corpus only has entity masks. This indicates that our model can use the the contextual knowledge provided by the retriever module even though it is not explicitly shown such knowledge during training.

Note that while REALM is similar to our proposed solution as far as the idea of retrieving relevant knowledge is concerned, the key difference in the two approaches lies in the source of knowledge being used. REALM relies on an unstructured text corpus (Wikipedia) as the source of knowledge and employs a computationally complex retrieve and read paradigm requiring additional training of the knowledge retriever model. Our proposed solution, on the other hand, uses structured knowledge which offers the

| Model | no. of params | Retrieval corpus size | Resources used |
|-------|---------------|-----------------------|----------------|
| Ours  | 110M          | 17M KB triples (∼100M words) | 8 GPUs  |
| REALM | 330M          | 5.5M documents (∼2B words) | 80 TPUs    |

Table 3: Resource requirements of our proposed approach and REALM.

advantage of being (almost) unambiguous and less resource-hungry compared to unstructured text. We present the resource requirements of our approach and REALM in Table 3. Note that the size of the external knowledge (in number of words) used by REALM is an order of magnitude greater, and requires three times the number of parameters compared to our model. Furthermore, REALM was trained for 200K steps with a batch size of 512 on an 80 TPU cluster, whereas our proposed solution is much more efficient being trained for 2K steps with a batch size of 256 on a machine with 8 Nvidia A100 GPUs. This computational efficiency of our proposed solution allows us to continue further work on improving our performance by enhancing the structured knowledge base and bridge the performance gap with more complex and computationally expensive models such as REALM.

### 3.3 Sensitivity to Contextual Signals in Input

PLMs are often sensitive to linguistic variations in the input and thus, can produce factually incorrect output. Poerner et al. [2020] have shown that the original BERT model is overly reliant on the surface form of entity names for making its predictions. For example, it will predict that a person with an Italian-sounding name was born in Italy even if this is factually incorrect. To evaluate the sensitivity and robustness of PLMs, they introduced LAMA-UHN (UnHelpfulNames), a much harder subset of LAMA where input probes with helpful entity names are removed and the PLM has little or no helpful contextual signals from other tokens in the probe. Table 4 reports the numbers for the two BERT variants and our solution. We report the aggregate numbers on the Google-RE and T-REx subsets as LAMA-UHN considers only these subsets of LAMA. While the outperformance of our solution on LAMA has already been established (Table 2), we observe that it significantly outperforms the BERT variants on the much harder LAMA-UHN probe too. Further, the decline in performance from LAMA to LAMA-UHN is only about $14\%$ for the proposed solution compared to almost a one-third drop in performance for both the BERT variants. Thus, our proposed solution offers significant and *robust* outperformance over PLMs.

### 3.4 Discussions

We now discuss some representative examples to illustrate the successes and failures of our proposed solution. Consider a test probe from the Google-RE subset of LAMA – *Nigel Pulsford plays ____*. Here, the correct output token is *guitar* and BERT model incorrectly predicts *sgt* as the output token. On the other hand, our proposed solution correctly predicts *guitar* as the answer to this probe. We argue that the knowledge retrieved by our Knowledge Retriever which includes the relevant fact *<Nigel Pulsford; has occupation; Guitarist>* has helped the model to produce the correct output. We present more such successful examples in the Appendix.

Next, we analyzed the cases where the proposed solution was producing incorrect output and observed three major categories. First, in a majority of the cases, the system failed to produce the correct answer because the required knowledge was not present in the knowledge base. This problem could be addressed by enhancing the knowledge base. Second, we observed that in a few cases, the required knowledge was not retrieved by the retriever despite being present in the knowledge base and in a small number of cases, the system made errors even after retrieving the relevant knowledge. This highlights the scope of improvement in our retrieval module as well as pre-training module, where further training could help the model make better use of the retrieved knowledge. Some representative examples of these different cases are presented in the Appendix.

Lastly, we noticed some errors that could be attributed to the characteristics of the LAMA probe. Specifically, there are input probes that refer to entities without providing any additional context for disambiguating the surface form. For example, the sentence "*James Johnson was born in ___*" has no clues to determine whether the prompt is referring to the basketball player, Virginia congressman, or the Governor of Georgia with this name. We also noticed certain probes where there are multiple correct completions and the benchmark considers only one of these as the correct answer. For example, "*Michelangelo is a ___ by profession*" can be correctly completed by *sculptor, poet, painter* or *architect*, but the evaluation considers only *poet* as the correct answer. Further, similar to the observation by Agarwal et al. [2021], we noticed some input examples with highly unambiguous language. For example, "*X died in ___*", can refer to either X's place of death or date of death but only the former is accepted as the correct answer. Lastly, there are cases where

slight (and correct) variations of the expected answer are evaluated as incorrect by the probe. For example, for the prompt "*Harashima is ___ citizen.*" *Japan* is provided as the correct answer while the prediction made by the proposed solution (*Japanese*) is considered incorrect.

## 4. Conclusion

We presented a simple way of augmenting PLMs with contextually relevant structured knowledge. Using Yago4 as our base knowledge base, we found that our proposed approach leads to significant performance improvements as measured by different subsets of the LAMA probe. We presented representative examples of cases where our proposed approach helps the language models as well as presented a thorough analysis of failures of our proposed model that helped us identify possible directions of future work.

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

# Appendix

| Input Query | Expected Answer | Model Output | Retrieved Candidates | Comments |
|---|---|---|---|---|
| Hans Gefors was born in ____. | Stockholm | Hamburg | (Hans Raj Hans birth date 1953-11-30) 
 (Hans Raj Hans given name Hans (name)) 
 (Hans Raj Hans nationality India) 
 (Hans Raj Hans has occupation Politician) 
 (Hans Raj Hans member of Indian National Congress) 
 **(Claus Gerson birth place Hamburg)** 
 (Hans Geister birth date 1928-09-28) 
 (Hans Gericke nationality Germany) | Corresponding fact not present in KB. We speculate that the candidate in bold led the model to predict Hamburg. Bert predicted *Oslo* as the answer. |
| Victor Salvi plays ____. | harp | quarterback | (Victor Salvi given name Victor (name)) 
 (Victor Salvi nationality United States) 
 (Victor Salvi death place Milan) 
 (Victor Salvi death date 2015-05-10) 
 (Victor Salvi birth place Chicago) 
 (Victor Salvi birth date 1920-03-04) 
 (Joan Lui actor Francesco Salvi) 
 (Victor Salvi birth date 1920-03-04) | Corresponding fact not present in KB |
| CBeebies is owned by ____. | BBC | Microsoft | (CBeebies founding date 2002) 
 (Gigglebiz creator CBeebies) 
 (CBEF contained in place Ontario) 
 (CBEF location Ontario) 
 (Bambi production company The Walt Disney Company) 
 (CBE Software founding date 2006) 
 (Paddington Bear (TV series) production company ITV Central) 
 (CBS Interactive parent organization CBS Corporation) | Corresponding fact not present in KB. |
| Ivan Petch was born in ____. | Concord | Sydney | (Ivan Petch birth date 1939-03-01) 
 **(Ivan Petch birth place Concord, New South Wales)** 
 (Ivan Petch family name Petch) 
 (Ivan Petch given name Ivan (name)) 
 (Ivan Petch nationality Australia) 
 (Ivan Petch has occupation Politician) 
 (Ivan Petch has occupation Electrical engineer) 
 (Ivan Petch alumni of Fort Street High School) | Correct fact is retrieved. However, the model is still not able to predict correct output. |
| Scientist was born in ____. | Kingston | London | (Scientist (musician) birth date 1960-04-18) 
 (Thomas Young (scientist) has occupation Physicist) 
 (I Am a Scientist date published 1994) 
 (Thomas Prince (scientist) has occupation Physicist) 
 (Bambi production company The Walt Disney Company) 
 (Allen Taylor (scientist) nationality United States) 
 (Lawrence Roberts (scientist) nationality United States) 
 (David Thomas (Canadian scientist) has occupation Biochemist) | Ambiguous query, leads to poor retrieval results. |

Table 5: Illustrative examples of cases where the proposed solution produced incorrect completions.

| | Relation | Input query | Our model prediction | BERT-base prediction | Candidates |
|---|---|---|---|---|---|
| **Google-RE** | birth-place | Stanley Corrsin was born in ____. | Philadelphia | London | (Stanley Corrsin birth date 1920-04-03)
(Stanley Corrsin nationality United States)
**(Stanley Corrsin birth place Philadelphia)**
(Stanley Corrsin given name Stanley (given name))
(Stanley Corrsin death date 1986-06-02)
(Stanley Corrsin has occupation Physicist)
(Stanley Corrsin alumni of University of Pennsylvania)
(Stanley Corrsin member of American Academy of Arts and Sciences) |
| | birth-date | Tom Coppola (born ____). | 1945 | 1975 | **(Tom Coppola birth date 1945-06-06)**
(Tom Coppola given name Tom (given name))
(Tom Coppola nationality United States)
(Tom Coppola family name Coppola (surname))
(Tom Coppola alumni of USC Thornton School of Music)
(Christopher Coppola birth date 1962-01-25)
(Anton Coppola nationality United States)
(Chris Coppola birth date 1962-01-25) |
| | death-place | Aglaja Orgeni died in ____. | Vienna | Bucharest | (Aglaja Orgeni death date 1926-03-15)
**(Aglaja Orgeni death place Vienna)**
(Aglaja Orgeni birth date 1841-12-17)
(Aglaja Orgeni birth place Rimavská Sobota)
(Aglaja Orgeni nationality Austria)
(Aglaja Orgeni nationality Hungary)
(Aglaja Orgeni has occupation Opera singer)
(Aglaja Orgeni death place Vienna) |
| **T-REx** | P106 | Cigoli is a ____ by profession. | architect | lawyer | **(Cigoli has occupation Architect)**
(Cigoli nationality Italy)
(Cino Cinelli has occupation Businessperson)
(Francesco Cirio has occupation Businessperson)
(Cigoli birth place Cigoli, San Miniato)
(Emilio Cigoli has occupation Stage actor)
(Francesco Cigalini has occupation Mathematician)
(Ciputra has occupation Businessperson) |
| | P463 | Phil Mogg is a member of ____. | UFO | parliament | **(Phil Mogg member of UFO (band))**
(Phil Mogg nationality United Kingdom)
(Phil Mogg birth date 1948-04-15)
(Phil Mogg birth place London)
(Mo Mozzali member of Minneapolis Millers)
(John Mogg, Baron Mogg nationality United Kingdom)
(Jamie Moyer member of Colorado Rockies)
(Jamie Moyer member of Philadelphia Phillies) |
| | P407 | Summerfolk was written in ____. | russian | english | (Summerland (novel) in language English language)
**(Summerfolk in language Russian language)**
(The World That Summer genre Neofolk)
(Summerfolk author Maxim Gorky)
(Summer (novel) in language English language)
(Summertime (novel) in language English language)
(A Summer Tale date published 2000)
(Summerteeth in language English language ) |
| | P1303 | Nigel Pulsford plays ____. | guitar | sgt | **(Nigel Pulsford has occupation Guitarist)**
(Nigel Pulsford given name Nigel)
(Nigel Pulsford birth date 1963-04-11)
(Nigel Pulsford nationality United Kingdom)
(Nigel Pulsford nationality Wales)
(Nigel Pulsford birth place Newport, Wales)
(William Pulsford nationality United Kingdom of Great Britain and Ireland)
(Reginald Purdell has occupation Actor) |

16

Table 6: Illustrative examples of cases where the proposed model successfully output the correct completions for various probes in LAMA.