# OpenReview forum: "No Need to Know Everything! Efficiently Augmenting Language Models With External Knowledge"
_AKBC.ws/2021/Workshop/CSKB — CSKB_

### Official Review · Reviewer_utnk · 2021-09-14
**Very well presented , clear and nice work**

**Rating:** 7
**Confidence:** 4

**Review:**

Even though large transformer-based pre-trained language models memorize factual knowledge storing large amounts of it in the parameters of the model is sub-optimal given the resource requirements and ever-growing amounts of knowledge. Instead of packing all the knowledge in the model parameters, the authors present an alternative is to provide contextually relevant structured knowledge to the model from an external knowledge base YAGO and train it to use that knowledge.

One of the really sensible and cool directions is masking named entities to test factuality instead of randomly sampling any word to mask as done by the original BERT paper. They mask named entities in an input sentence and then retrieve relevant tuples from Yago that can complement the input. What is even better is the authors use a product of triplet and relation embedding with the input sentence to make the reranking step efficient. The external knowledge base tuples coupled with the input help the model learn better association ( For example "born in" can be related to PlaceOfBirth). The authors modify the MLM objective in the sense that along with the input they also condition it on top k retrieved knowledge-base tuples separated by [SEP] tokens. I personally loved the diagram as it tells the story in one picture.

Finally, while the performance is behind Google's REALM model, it is indeed impressive how better it is compared to vanilla bert base or bert-large model for LAMA probe. This is a timely important work given the fact that they achieve this with much much lesser resource and kind of cool that now we can make language models adapt to newer facts by just aligning them to a knowledge base with recent facts.

The idea is not something out of the box and can be thought of by many people in the community. But I don't feel the paper is worth nitpicking on the realm of novelty. The claims are well proven empirically. The Paper is well written, presented and if the code is released including the yago4 integration it could be of much potential help to the community

---

### Decision · Program_Chairs · 2021-09-18

Accept